# Global warming may increase the burden of obstructive sleep apnea

Bastien Lechat ●[1] ✉, Jack Manners[1], Lucía Pinilla ●[1], Amy C. Reynolds[1], Hannah Scott ●[1], Daniel Vena[2], Sebastien Bailly ●[3], Josh Fitton[1], Barbara Toson[1], Billingsley Kaambwa[4], Robert J. Adams[1], Jean-Louis Pepin ●[3], Pierre Escourrou[5], Peter Catcheside ●[1] & Danny J. Eckert ●[1]

High ambient temperatures are associated with reduced sleep duration and quality, but effects on obstructive sleep apnea (OSA) severity are unknown. Here we quantify the effect of 24 h ambient temperature on nightly OSA severity in 116,620 users of a Food and Drug Administration-cleared nearable over 3.5 years. Wellbeing and productivity OSA burden for different levels of global warming were estimated. Globally, higher temperatures (99[th] vs. 25[th]; 27.3 vs. 6.4 °C) were associated with a 45% higher probability of having OSA on a given night (mean [95% confidence interval]; 1.45 [1.44, 1.47]). Warming-related increase in OSA prevalence in 2023 was estimated to be associated with a loss of 788,198 (489,226, 1,087,170) healthy life years (in 29 countries), and a workplace productivity loss of 30 (21 to 40) billion United States dollars. Scenarios with projected temperatures ≥1.8 °C above pre-industrial levels would incur a further 1.2 to 3-fold increase in OSA burden by 2100.

In 2023, mean temperature recordings in the northern hemisphere were the highest on record in over 2000 years[1], with temperatures 2.07 °C warmer than the pre-industrial period (~1850-1900)[1]. Mean global temperature is projected to increase by 2.1 to 3.4 °C above pre-industrial levels by the end of the century, in the absence of additional reductions in greenhouse gas emissions[2]. High ambient temperatures have strong, well-established, negative effects on health[3,4], including mental health outcomes[5–7], reduced physical activity[8], and ultimately, mortality[9–14]. High ambient temperatures are also associated with considerable reductions in sleep duration and quality[15–17], including a near doubling of short sleep (<6 h) prevalence[18]. Sleep loss due to global warming in 2023 was recently estimated to incur a loss of 3.9 million years of life because of disability or death[18].

Obstructive sleep apnea (OSA) is the most common sleep-related breathing disorder with an estimated global prevalence of nearly 1 billion adults[19]. OSA severity is typically quantified using the apnea-hypopnea index (AHI), a count of partial (hypopnea) or total (apnea) upper airway collapses per hour of sleep. Untreated OSA is associated with a wide range of adverse health outcomes[20], road safety events[21], and all-cause mortality[20]. Increased OSA severity also has a dose-response relationship with the incidence of co-morbid chronic conditions, such as hypertension and type-2 diabetes[20]. In addition, untreated OSA is associated with large decreases in workplace productivity and absenteeism. This is estimated to cost the Australian economy nearly 4 billion AUD[22] and the US economy nearly $87 billion US per year[23]; while data for other countries is lacking.

Some studies have shown an association between outdoor ambient temperature and OSA severity[24–27]. A major limitation is that most prior studies are cross-sectional and use single-night estimates of AHI, therefore neglecting to account for the substantial night-to-night variability in AHI[28–30]. Furthermore, to the best of our knowledge, personal, and social burden of OSA due to increased temperatures, including changes in prevalence, wellbeing burden, and workplace productivity loss have not been quantified to date. Given the potential future impact that global warming is expected to have on OSA prevalence, better estimates of the subsequent burden are needed.

[1]Adelaide Institute for Sleep Health and FHMRI Sleep Health, College of Medicine and Public Health, Flinders University, Adelaide, Australia. [2]Division of Sleep and Circadian Disorders, Brigham and Women's Hospital and Harvard Medical School, Boston, MA, USA. [3]Univ. Grenoble Alpes, HP2 Laboratory, Inserm U-1300, CHU Grenoble Alpes, 38043 Grenoble, France. [4]Health Economics Unit, College of Medicine and Public Health, Flinders University, Health Sciences Building, Sturt Road, Bedford Park, SA 5042, Australia. [5]Centre Interdisciplinaire du Sommeil, Paris, France. ✉e-mail: bastien.lechat@flinders.edu.au

Accordingly, in this global study involving 116,620 users of a validated and Food and Drug Administration (FDA)-cleared nearable OSA monitor, we investigate the dose-response relationship between daily ambient temperature and nightly OSA status, using data from the fifth generation of European Reanalysis (ERA5) dataset[31]. We demonstrate that high-temperature days (99th percentile; 27.3 °C) increase the probability of OSA by 45% compared to low-temperature days (25th percentile; 6.4 °C). The effect size varies by country and is generally stronger in European countries. The increased prevalence of OSA attributed to global warming in 2023 is associated with a loss of over 780,000 healthy life years and 105 million workplace productivity days across 29 countries, resulting in an estimated economic cost of ~98 billion United States dollars. Projections estimate that a ≥ 1.8 °C increase in global ambient temperature above pre-industrial levels could double the burden, impacts, and costs of OSA by 2100.

## Results

Data for final analysis was acquired from 116,620 regular users of the under-mattress OSA sensor, for a combined total of ~62 million nights. The most common reason for excluding participants (N = 8261, ~6.6%) was a lack of regular use of the device (see flow chart in Supplementary Fig. 1). Users were mainly middle-aged and predominantly male (22.7% female), with a median [IQR] of 509 [246, 787] sleep recordings per user (Table 1). The overall prevalence of moderate-to-severe (AHI ≥ 15 events/h) and severe OSA (AHI ≥ 30 events/h) was 25.4% and 8.9%, respectively. Users were located across most world regions (Fig. 1), including 41 countries with at least 100 users. OSA prevalence was between 15 and 32% depending on the country (Fig. 1), and demographics were consistent across countries (Table S1).

### Ambient temperature and OSA prevalence

There was a non-linear dose-response association between 24 h daily average ambient temperatures and the risk of having OSA on the following night (Fig. 2a). Globally, the probability of having OSA was 45% (risk ratio (RR) [95%CI]; 1.45 [1.44, 1.47]) higher during days with high temperatures (99th; 27.3 °C) vs. during days with lower temperatures (25th; 6.4 °C). Similarly, the probability of having severe OSA was 49% (1.49 [1.46, 1.52] higher in high vs. low temperature days. The probability of nightly OSA was higher at T99 in countries with lower gross domestic product (GDP) per capita (Fig. 2b). Out of the 41 studied countries, we found a significant association between ambient temperature and increased probability of nightly OSA in 29 countries (Fig. 2d). The effect size at T99 varied by country (Fig. 2c) and was generally stronger in European countries, with up to ~2.0 -fold increase in the probability of nightly OSA. Similar results were observed for the probability of nightly severe OSA (Table S1). The number of users was generally lower in countries where we did not find a significant

association between temperature and OSA prevalence (Table S1). The association between ambient temperature and the probability of nightly OSA was stronger in males, users with higher body mass index (BMI), and among participants with usual sleep duration longer than 6 h (Supplementary Figs. 2 and 4). Age did not have a strong effect on the association between temperature and nightly OSA (Supplementary Figs. 2 and 3). Heterogeneity in country demographics (see Table S1), including mean age (p-value = 0.17), mean BMI (p-value = 0.57), and sex proportion (p-value = 0.68) was not associated with a differential effect in the association between ambient temperature and the probability of nightly OSA. The exposure-response curve between ambient temperature and the probability of nightly OSA remained largely similar in the sensitivity analyses, when minimum or maximum 24 h temperature were used as exposure variables (Supplementary Fig. 4). The exposure-response curve between ambient temperature and the probability of nightly OSA also remained similar when adjusting for particulate matter with aerodynamic diameter <10 μm (Supplementary Fig. 5), previously shown to be associated with increased OSA severity in some studies[26]. The association between temperature and nightly OSA was also not modified by air pollution levels (Supplementary Fig. 5). A total of 9.4% of nightly sleep recordings did not have an AHI measurement, as the AHI cannot be calculated for nights with a sleep duration of less than 5 h. There was a bias towards a lower effect size in the association between temperature and nightly OSA for participants with a higher amount of missing AHI data (see Supplementary Fig. 6). Given the association between high temperature and short sleep duration[16,18], this may suggest that our estimates are conservative.

### Wellbeing burden of warming-related increase in OSA prevalence

We use disability adjusted life years (DALYs), a standardized measure employed by the Global Burden of Disease (GBD)[32] studies that captures the combined impact of illness, injury, and premature mortality, to quantify the wellbeing and societal burden due to increased prevalence of OSA from high temperatures. DALYs combines time lost due to premature death (years of life lost; YLL) and time lived with less than-optimal health (years lived with disability; YLD). The increased prevalence of OSA due to global warming was associated with a strong societal impact. For example, in the UK (London; Fig. 3a), we observed a ~ 0.7 °C temperature increase from 2000 to 2023, which corresponded with an additional ~150 million person-days with OSA in 2023 compared to 2000. The increase in OSA prevalence due to global warming was associated with a loss of 59,365 DALYs (uncertainty range: 36,975 to 81,755), including 58,341 YLDs (36,463 to 80,2189), and 1024 YLLs (512 to 1,536); 92% higher than in 2000 (Table S2). In the 29 countries where we found an association between OSA prevalence and ambient temperature (involving 1.02 billion people), temperature-

## Table 1 | Demographics information of the user's sample

| | | Overall | OSA severity category | | | |
| | | | No OSA | Mild | Moderate | Severe |
|---|---|---|---|---|---|---|
| n | | 116,620 | 49,301 | 37,696 | 19,248 | 10,375 |
| Age, years | | 49 (14) | 43 (12) | 50 (13) | 56 (13) | 58 (13) |
| Sex, n (%) | Male | 90140 (77.3%) | 35058 (71.1%) | 29781 (79.0%) | 16146 (83.9%) | 9155 (88.2%) |
| | Female | 26480 (22.7%) | 14243 (28.9%) | 7915 (21.0%) | 3102 (16.1%) | 1220 (11.8%) |
| Body mass index, kg/m² | | 27.6 (5.6) | 25.8 (5.0) | 28.0 (5.3) | 29.5 (5.5) | 31.5 (6.4) |
| Number of recordings, Median [IQR] | | 509 [246,787] | 501 [242,772] | 525 [261,800] | 519 [252,802] | 480 [211,781] |
| Total sleep time, h | | 7.4 (0.8) | 7.4 (0.7) | 7.4 (0.8) | 7.3 (0.9) | 6.9 (1.2) |
| Mean apnea-hypopnea index, events/h | | 11.3 (13.5) | 2.0 (1.5) | 9.2 (2.8) | 21.1 (4.2) | 45.4 (14.2) |

Obstructive sleep apnea (OSA) severity categories were based on the apnea-hypopnea index using the following cut-offs: mild: 5–15; moderate: 15–30 and severe: ≥30 events/h.

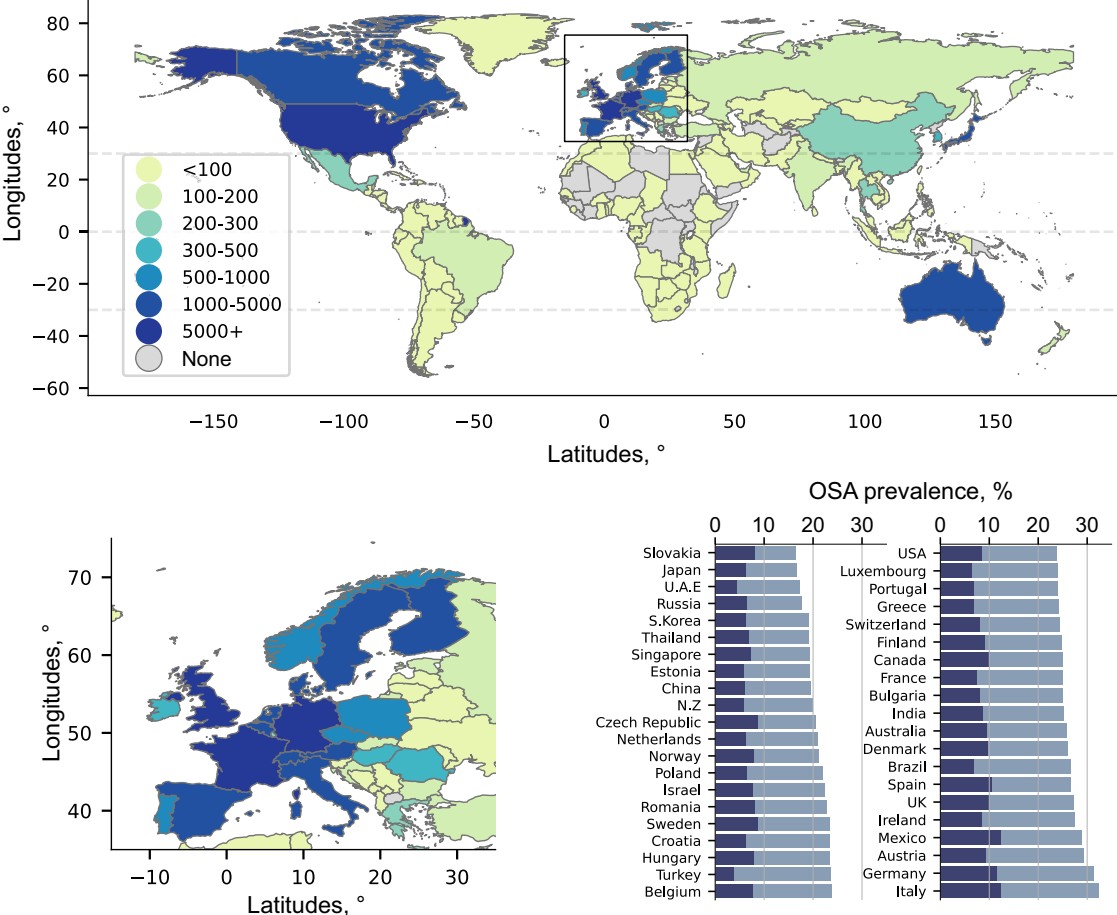

**Fig. 1 | Geographical location of the under-mattress regular user's sample.** Country by country prevalence of moderate-to-severe obstructive sleep apnea (OSA) (blue) and severe OSA (dark blue) is highlighted in the bar chart. Source data are provided as a Source Data file.

related increase in OSA prevalence was associated with a loss of 788,198 (489,226 to 1,087,170) DALYs in 2023, including 27,180 (13,590 to 40,770) years of life lost. The economic cost of this wellbeing impact in 2023 was estimated to be ~68 billion USD (Table S2) in the 29 studied countries. These estimates were ~2-fold higher than in 2000.

We used the climate scenarios CMIP6[33] from the 2021 sixth assessment report of the IPCC[2]−termed the Shared Socioeconomic Pathway (SSP), to estimate future OSA burden for different levels of global warming. We downloaded daily temperatures from 2023 to 2100 projections for 4 SSPs (SSP126, SSP245, SSP370, SSP585), based on 27 climate models from multiple countries and climate modelling groups. In the 29 studied countries, a ~ 1.8°C increase in global ambient temperatures above pre-industrial levels (SSP126 by 2100) was associated with in a ~15% increase in wellbeing burden compared to 2023 estimates (Table S3). Scenarios with projected increased temperature higher than SSP126 were estimated to further increase the wellbeing burden by 10 to 30% by 2050, and 1.5 to 2.5-fold by 2100, depending on the country (Fig. 3b). Temperatures higher than 1.8 °C above pre-industrial levels were estimated to incur an additional 12 to 36 million YLDs, 0.5 to 1.3 million YLLs, and an associated cost of 1.0 −2.8 trillion USD, over the 2023 to 2100 period. Country by country rates of DALYs per 100,000, absolute number of YLLs, absolute numbers of YLDs, and associated economic cost persons under different scenarios in 2050 and 2100 are presented in Tables S3 to S6. Accounting for population forecast and a potential 30% increase in OSA prevalence by 2050 due to rise in obesity[34] would further increase these estimates in most countries by 10 to 25%. Comparison of the main model estimates vs. the alternative scenario is available in Tables S7 and S8. Under these

assumptions, any other scenario than SSP126 would be estimated to incur an additional 16.9 to 47.4 million DALYs lost over the 2023 to 2100 period.

## Workplace productivity burden of warming-related increase in OSA prevalence

OSA has a considerable impact on productivity and is associated with increased likelihood of absenteeism and presenteeism (being at work but not functioning at full capacity)[22,35]. The calculated increase in OSA prevalence from global warming is estimated to incur additional productivity losses (Fig. 4). In 2023, for the 29 studied countries, the increase in OSA prevalence due to higher temperatures was associated with an additional 80 million presenteeism days (59 to 101 million) and an additional 25 million absenteeism days (12 to 38 million). The total economic cost of this loss of labor was 30 (21 to 40) billion USD. Globally, the rate of labor loss in 2023 was 8.4 (5.7 to 11.1) million USD per 100,000 persons and was double the rate of labor loss of 2000 (4.0 [2.7 to 5.2] million USD; see Table S7).

Scenarios with projected increased temperatures higher than 1.8 °C (SSP126) above pre-industrial levels are expected to increase labor loss by 10 to 25% by 2050, and 1.3- to 2.2-fold by 2100, depending on the country (Fig. 4; Tables S10, S11). These scenarios are projected to incur an additional cost of 0.9 to 2.0 trillion USD in labor loss over the 2023 to 2100 period (Fig. 4). In 2100, the labor loss rates per year for different climate models in the 29 studied countries are projected to be 37 billion (10.0 million per 100,000 - SSP126), 50 billion (13.3 million per 100,000 - SSP245), 69 billion (18.4 million per 100,000 - SSP370) and 84 billion (22.8 million per 100,000 - SSP545), see

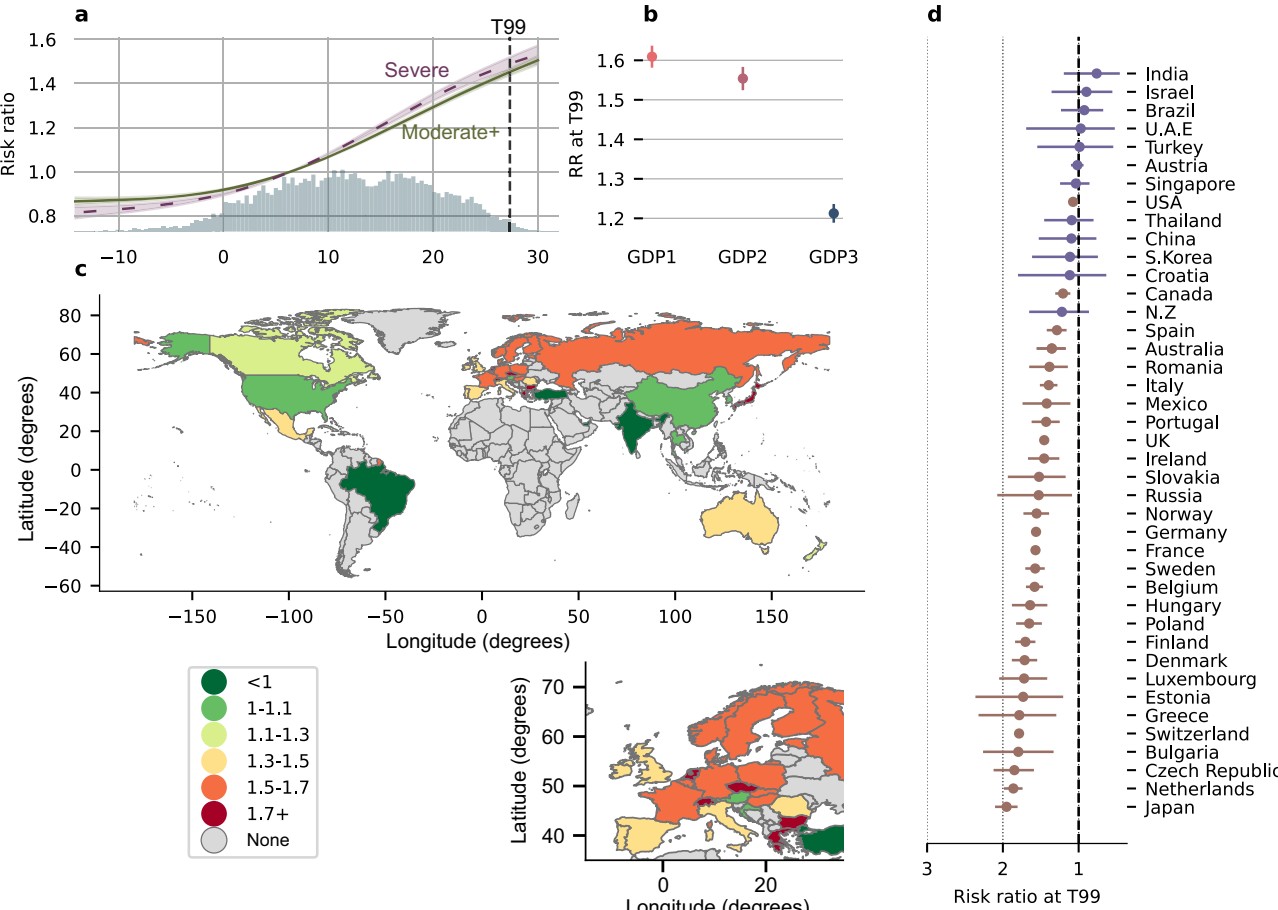

**Fig. 2 | Cumulative exposure–response between ambient temperature and nightly obstructive sleep apnea (OSA) status per location. a** Exposure-response curves between temperature and the risk ratio (RR) for nightly OSA (solid green) or nightly severe OSA (dashed purple). Shaded area represents 95%CI. Distribution of temperature (histogram) and temperature at the 99th percentile (T99; dashed black line) are also highlighted. **b** Subgroup analyses including the mean RR (and 95%CI) for nightly OSA at T99 based on the country of residence's gross domestic product (GDP) per capita (tertiles – N, tertile 1 :40152; tertile 2: 37722; tertile3: 38468). **c** RR for nightly OSA at T99 by geographical location. **d** Country by country analysis with mean RR (and 95%CI) for nightly OSA at 99th vs. 25th percentile of temperature (brown indicates *p*-values < 0.05 and purple *p*-values > 0.05). Exact N and RR (95% CI) for each country is available in Table S1. Source data are provided as a Source Data file.

Tables S10 and S11. Country by country rates and absolute numbers of labor loss for different climate scenarios in 2050 and 2100 are presented in Tables S10 and S11.

## Discussion

High ambient temperatures (99th vs. 25th; 27.3 vs. 6.4 °C) are associated with a ~ 45% increase in the probability of having OSA on a given night, globally. The increase in OSA prevalence in 2023 due to global warming was associated with a loss of 788,198 (489,226 to 1,087,170) healthy life years, and a productivity loss of ~105 million days from impaired workplace attendance in the 29 studied countries. In 2023 alone, the estimated total economic cost associated with the warming-related increase in OSA prevalence was ~98 billion USD, including 68 (34 to 102) billion USD from additional wellbeing burden and 30 (21 to 40) billion USD from workplace productivity loss.

Modelled temperatures consistent with the continuation of government climate change policies implemented by 2020 are projected to lead to global warming of 2.1 to 3.4 °C by 2100[2,36]. The health and economic impact of these estimates would be consequential, and the increase in OSA prevalence due to rising temperatures in such scenario may double the overall OSA burden. Indeed, rates of DALYs, without considering temperature, for OSA in 2019 in Australia were already ~200 to 260 DALYs per 100,000[22]. Estimated 2100 rates of DALYs observed for most countries under SSP245 (~2.8 °C degrees of

warming) were also around 150 to 300 DALYs per 100,000 people depending on the country, which would approximately double the current OSA burden. A previous, not yet peer reviewed, study also estimated a 122 billion USD wellbeing cost due to warming-related increase in short sleep prevalence[18]. Together, these studies highlight the economic burden associated with sleep-related inadequacy from rising temperatures. In addition, these estimates contribute to the growing body of evidence that investment to limit global warming would be a cost-effective strategy to prevent climate-related sleep impairment and its consequences[37,38].

The sample used in this study likely under-represents lower socioeconomic groups, given that all participants owned a consumer-grade sleep tracking device, and most participants resided in highly developed countries. Hence, participants may have also had access to more favorable sleeping environments and heat stress-mitigation strategies such as air conditioning, which are less readily available in lower socioeconomic populations[39]. There are large global inequalities in availability of population sleep health data—with only 22% WHO member states (mostly highly developed) having published population data in sleep duration[40]. Similarly, published global estimates of OSA prevalence are derived from population studies in mostly high socio-economic countries[19]. Our study is also biased towards higher socioeconomic countries and highlights the urgent need for global strategies to collect appropriate sleep and temperature data

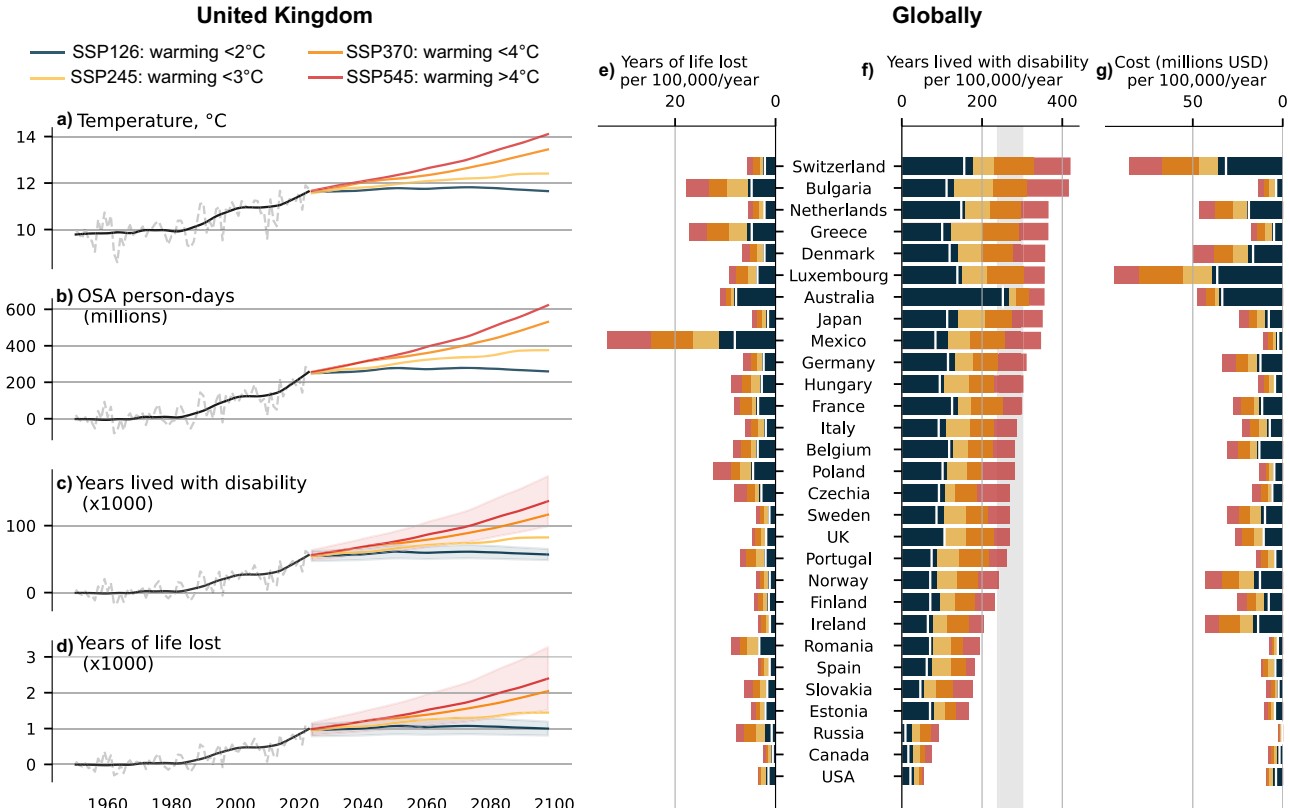

**Fig. 3 | Projected wellbeing burden of warming-related increase in moderate-to-severe obstructive sleep apnea (OSA) prevalence.** Wellbeing burden of warming-related increase in moderate-to-severe OSA prevalence is exemplified for the UK (**a**–**c** and **d**) and globally (**e**–**g**). **a** Historical and projected temperatures for different climate scenarios from the Intergovernmental Panel on Climate Change (IPCC) were used to (**b**) derive the number of additional person-days with OSA due to temperature increases, which were subsequently used to calculate (**c**) years lived with disability (YLD) and (**d**) years of life lost (YLL). Baseline temperatures were based on the historical 1950-1990 data. The wellbeing and economic impact of the estimated temperature-related increase in OSA by 2100 for different climate scenarios from the IPCC were calculated for (**e**) YLL, (**f**) YLD, and (**g**) wellbeing economic cost, expressed as rate per 100,000 persons per year. The white line-marker represents current rates (as of 2023) of YLL, YLD, and economic cost. Shaded area in the YLD graph represents the yearly wellbeing burden associated with OSA based on Australian estimates[22]. USD US dollars, SSP Shared Socioeconomic Pathway. Source data are provided as a Source Data file.

worldwide[40]. In addition, 77% of the participants in the current study were male. We did not detect a large difference between sex in the temperature-OSA association. However, this study sample may not be representative of the wider general population and does not resolve the long-standing issue in sleep research of under-representation of female in such analyses.

Furthermore, since we lacked access to precise geographical locations and the indoor temperature of users, exact temperature-related effects could not be calculated. The potential confounding effect of air pollution on the association between high temperature and nightly OSA status may also have been underestimated since we lacked precise geographical locations and indoor measurements of air pollution. We also lacked clinical information on participants' co-morbidities, treatment status, daytime symptoms, and impairments, and thus, could not explore potential modifiable effects of these conditions and symptoms on the detected association. Individuals with OSA and daytime sleepiness are at much greater risk of motor-vehicle accidents compared to those with OSA alone[41]. Hence, the temperature-related burden of OSA may be greater in this specific population. This aligns with previous studies that indicate that sleep loss and ambient temperature disproportionately affect more vulnerable populations[16,17]. High ambient temperatures have also been found to exacerbate the severity of chronic diseases[42]. Thus, further consideration of co-morbidities and daytime symptoms associated with OSA is warranted to better determine the impact of ambient temperature on OSA in vulnerable populations, and the additional burden

this may incur. Although our robust time-series analysis design provides support for a potential causal association between high temperature exposure and increased OSA prevalence, other time-varying confounders (e.g., alcohol intake) were not available in the current study, and should be accounted for in further studies.

The under-mattress sensor used in this study does not measure the contribution of some physiological aspects of OSA (e.g., hypoxia, arousals) which may undermine OSA detection. However, existing validation studies versus gold standard polysomnography in over 150 participants[28,43] support the device performance characteristics. Furthermore, OSA prevalence estimates using non-contact multi-night data found in this study yield very similar findings to previously published literature[19,28]. Similarly, previous studies that used this device have shown that misclassification rates and AHI variability are comparable to other devices[30], and that the effect size of the association between the estimated AHI and health outcomes is also similar to existing epidemiological trials[44,45]. Furthermore, conventional single-night polysomnography can lead to inaccurate OSA diagnoses and severity estimation in 20 to 50% of patients due to high-night-to-night variability of OSA[28,29,46]. Thus, these studies provide support that the multi-night AHI estimates used in the current study provide comparable or superior insight to conventional single-night polysomnography AHI while being less cumbersome and allowing nightly data collection over ~1.5 years per individual.

Our main analysis aimed to model rising temperature's impact on OSA-related wellbeing and productivity by holding other factors

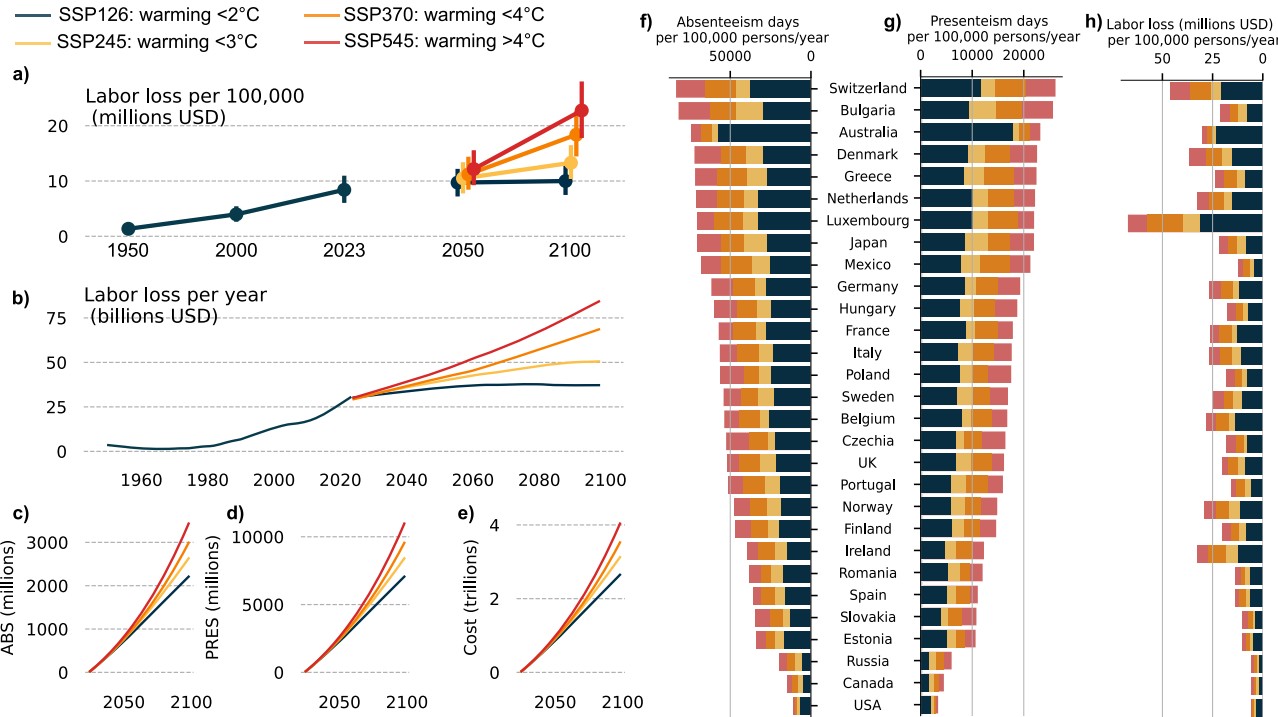

**Fig. 4 | Global projected workplace productivity loss from warming-related increase in moderate-to-severe obstructive sleep apnea (OSA) prevalence.** Calculated labor loss (in USD) for different climate scenarios from the Intergovernmental Panel on Climate Change (IPCC) (top) with different levels of future global warming over the 29 studied countries per (**a**) 100,000 people (mean and 95%CI across countries); (**b**) year. Cumulative sum of (**c**) absenteeism days, (**d**) presenteeism days, and (**e**) associated economic cost for different climate scenarios over the 2023–2100 period. Rates of (**f**) absenteeism (**g**) presenteeism, and (**h**) labor loss in 2100 per 100,000 people under different climate scenarios for the 29 studied countries. USD: US dollars; SSP: Shared Socioeconomic Pathway. Source data are provided as a Source Data file.

constant. However, BMI is projected to increase over the next 50 years, likely leading to higher OSA prevalence[34]. This suggests our main estimates may understate the future temperature-related OSA burden. Indeed, once accounting for BMI-related increase in OSA prevalence in our sensitivity analyses, our estimates were 10 to 30% higher. On the other hand, recent developments in OSA treatment[47], including FDA-approved weight loss medications[48], could help manage OSA more effectively. Given that approximately 80% of OSA patients are currently undiagnosed or untreated[49], improved treatment rates could serve as an adaptation mechanism to offset temperature-induced increases in OSA burden. We also assumed constant heat adaptation at the population level. However, heat-related mortality in Europe has declined over the past two decades[50], and global air conditioning adoption has increased from 19.3% (2000) to 35.3% (2021). Studies that have investigated heat adaptation strategies and its effects on sleep and OSA are scarce[15]. Thus, we could not account for heat adaptation. In addition, heat adaptation is not uniform: highly developed countries have 44-48% household air conditioning coverage, while low and medium developed countries have 4% and 14%, respectively[51]. Accordingly, given our sample's socioeconomic bias toward developed nations, we likely underestimate the global impact of temperature-induced OSA, even without accounting for increased adaptation capabilities.

In conclusion, our study underscores the potential significant impact of increasing ambient temperatures on the prevalence of OSA, globally. Without substantially greater policy change to slow global warming, the health and economic burdens associated with OSA may double by 2100. These results highlight the urgency of limiting global warming to 1.5 °C above pre-industrial levels, in alignment with the Paris Agreement. Our findings also emphasize the immediate need for targeted measures to potentially minimize the health and economic impacts of the growing OSA prevalence associated with rising temperatures.

## Methods

The current study was approved by the Flinders University Human Research Ethics Committee (Project number: 4291). This study complies with the recommendations of the GATHER statement[52], and GATHER checklist can be found at the end of supplementary materials.

### Participants

Retrospective data were acquired from 125,295 participants who registered to use an under-mattress sleep sensor (Withings Sleep Analyzer/Sleep Rx; WSA) between January 2020 and September 2023. To be included in this analysis, participants were required to have ≥28 nights of data and an average of ≥4 sleep recordings per week. When participants sign-up to the Withings app, they are prompted to fill in their age, sex, height, and weight. Information on gender is not collected. Participants were required to be 18 years or older to be included in this study. Participants were geo-localized to the closest city within a given time zone. A more precise location was not available due to concern for re-identification and privacy. All participants provided written consent through the Withings app for their deidentified data to be used for research purposes when signing up for a Withings account.

### Country level information

Country level GDP per capita and GDP per capita employed were collected from the World Bank (https://data.worldbank.org/indicator/NY.GDP.PCAP.CD). Population and life expectancy estimates were based on the United Nations World 2024 Populations Prospects report[53].

## Objective OSA monitoring

The WSA is a FDA-cleared sleep monitoring device placed under the mattress that estimates sleep timing, quality, and breathing using ballistography and sound data (FDA 510(k) number: K231667)[43]. WSA-derived sleep duration, timing, and AHI have good agreement against polysomnography in multiple independent validation studies[28,43,54]. The WSA-estimated AHI also shows good agreement with in-laboratory polysomnography-derived AHI to classify moderate-to-severe OSA (88% sensitivity and 88% specificity to detect ≥15 events/h during sleep)[28,43]. The AHI cannot be reliably calculated for nights with a sleep duration of less than five hours[43], and these nights were excluded from analysis. Our primary outcome of interest was the nightly OSA status for a given participant on a given night, based on the AHI and clinically defined severity categories (nightly OSA: AHI ≥ 15; and nightly severe OSA: AHI ≥ 30 event/h). For each participant, we also calculated the average OSA severity category based on the average AHI on all available nights and using pre-defined clinical cut-offs (mild: 5–15; moderate: 15–30 and severe: ≥30 events/h). The prevalence of OSA per country was calculated based on the average OSA severity category for each participant.

## Assessment of weather information and climate projections

Weather information was time-matched for each day of data for every user based on their closest location, collected from the Copernicus Climate Change Service[55]. We extracted and calculated 24 h minimum, maximum, and average daily temperature as well as total cloud cover, surface pressure, and relative humidity from the fifth generation of ERA5 dataset[31]. We extracted fine particulate matter (aerodynamic diameter <2.5 μm) concentration from the ECMWF Atmospheric Composition Reanalysis 4 model as a measure of air quality[56]. These re-analysis models have been validated against station-based measurements[57–59] and are widely applied in environmental health research[57,58,60]. We used the climate scenarios CMIP6[33] from the 2021 sixth assessment report of the IPCC[2]. We downloaded daily temperatures from 2023 to 2100 projections for 4 SSPs (SSP126, SSP245, SSP370, SSP585), based on 27 climate models from multiple countries and climate modelling groups. Averaged daily projections were calculated as the daily median of the 27 projections. Detailed information on pre-processing and weather variable extraction can be found in the Supplementary Methods.

## Statistical analysis

Non-linear fixed effect models were used to study the effects of 24 h average ambient temperature on nightly OSA status on the following night, defined as an AHI ≥ 15 events/h for that night. We followed the recently proposed case-time series methodology[61] which has been used extensively in environmental health studies[42,62–65], including in our previous study estimating the effect of high ambient temperature on sleep[18]. We used participant/year/month strata intercepts, natural splines of time (day of the year; with 4 degrees of freedom [df]), and indicators of the day of the week to model individual baseline risks, in addition to shared long-term, seasonal, and weekly trends. Models were further adjusted for total cloud cover (4 df), relative humidity (4 df), density of particulate matter with aerodynamic diameter <2.5 μg/m³ (4 df), surface pressure (4 df), total precipitation (4 df), wind speed (4 df) and variation in total sleep time (4 df) as these potential confounders may have effects on OSA prevalence within the 1-month stratum[27].

The exposure-response association of interest between temperature and nightly OSA status was modelled using distributed lags[9,66]. We primarily chose a 4-day lag structure since the effect of temperature on sleep duration was shown to be significant up to the fourth day[18]. Other lag structures were explored but made little difference to the main findings (see Supplementary Methods and Supplementary Figs. 7 and 8). Similar to previous studies on the effect of temperature on health[9,10,13,66], we used the sum of the effect over the

lag period to summarize the association between temperature and nightly OSA status. This analytic approach was conducted on the complete dataset, and in independent analyses stratified by country. Further analyses examined specific subgroups of interest, including age groups (10-year categories), sex, BMI categories, habitual sleep duration categories (average sleep duration over the recording period), and tertiles of GDP per capita. The above analyses also estimated the association between ambient temperature and severe OSA (AHI ≥ 30 events/h) prevalence rates. Models are summarized using RR [95%CI], calculated from the odds ratio using existing formula[67]. We used the R programming language (version 4.3.3), specifically with the *dlnm*[68] (version 2.4.7) and *gnm*[69] packages (version 1.1-5). Sensitivity analyses were conducted, and the analysis was repeated by using minimum 24h-average temperature and maximum 24h-average temperature to further validate our findings. We also conducted a sensitivity analysis to investigate potential bias due to missing data when users are sleeping less than five hours. In this sensitivity analysis, we calculated the exposure-response curve between temperature and nightly OSA for each quartile of missing data proportion (determined per-participant).

## Calculation of the global warming-related increase in nightly OSA prevalence

For each country, we calculated the difference in temperature for a given year compared to the 1950–1990 historical average temperature (see Fig. 5a for an example in the UK). This difference in temperature was used to calculate the change in the probability of having OSA between a given calendar year (e.g., 2022) versus historical data, which was subsequently transformed into a RR. Using this RR, we calculated the relative change in nightly OSA prevalence associated with global warming-related increased temperatures for each calendar day (in person-days) for that specific year and country. This approach was repeated for each location and each calendar day between 1950 and 2100, as well as for each tested climate projection.

## Wellbeing burden modelling

DALYs are a standardized summary measure employed by the World Health Organization and the GBD studies[32] to quantify the impact of illness, injury, and/or dying prematurely. DALY combines time lost due to premature death (YLL) and time lived with less than-optimal health (YLD). DALYs measures have been used to assess the effect of temperature on health in other studies[70,71]. For this study, we followed the methodology previously used to assess the wellbeing burden of OSA in Australia[22,72] and our previous report on the global wellbeing burden of reduced sleep duration[18]. DALYs, YLLs, and YLDs were quantified using the calculated warming-related person-days increase in OSA prevalence (see previous section) and published literature on associations between OSA and health outcomes.

Given the well-established association between OSA and motor-vehicle accidents, we estimated the years of life lost (YLL) due to increased motor vehicle accidents associated with the warming-related increase in OSA prevalence. We also estimated temperature-related OSA effects on other conditions that may induce less-than-optimal health and hence increase the number of YLD. In our study, we based these estimates for the YLD using data from a previous studies[22,72]. More formally, $YLL_{c,y}$ for a given country $c$ at a year $y$ was calculated as follows:

$$YLL_{c,y} = \frac{1}{365} \sum_{d=1}^{365} [RR_c(T_{d,y}) - RR_c(Thist_{d,1950-90})] * p0_c \tag{1}$$
$$* APOP_{c,y} * MVA_{OSA\_death\_rate_c} * RLE_c, \text{ with,}$$

$$MVA_{OSA\_death\_rate_c} = MVA_{death\_rate_c} * RR_{MVA\_OSA}, \tag{2}$$

**a)**

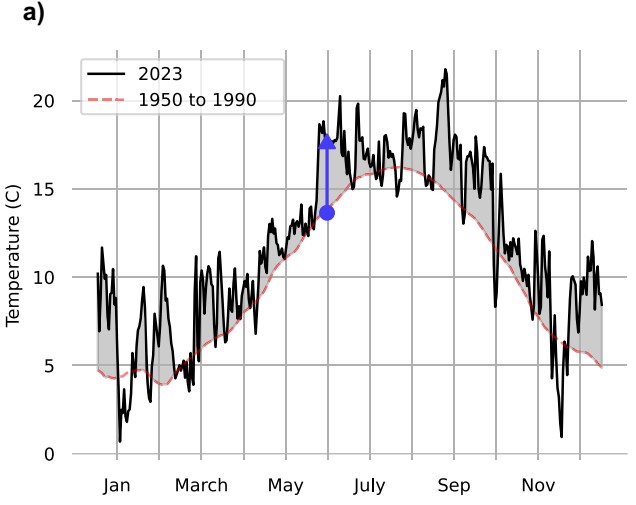

**b)**

**Fig. 5 | Example of methodological calculation of the warming-related increase in obstructive sleep apnea (OSA) prevalence for the UK. a** The difference in temperature for a given year (here 2023−black solid line) was compared to the 1950−1990 historical average temperature (dashed red line). **b** This difference in temperature (blue arrow) was used to calculate the change in the risk of nightly OSA between current vs. historical average temperature, for a particular day. This daily change was summed over the year and used to calculate the associated wellbeing and workplace productivity burden (see text for further details). Figure 5b represents the cumulative exposure−response (and 95%CI in shaded area) between ambient temperature and nightly OSA in the UK. Source data are provided as a Source Data file.

where $MVA_{death\_rate_c}$ is the daily rate of fatal motor vehicle accidents for a given country $c$, extracted from the GBD[73]; $RR_{MVA\_OSA}$ is the daily RR of fatal motor vehicle accidents associated with OSA, here estimated to be 100% based on previous literature[21,41,74–76]; $RR_c$ is the daily RR function for OSA function for a given country $c$; $Thist_{d,1950−90}$ is the average historical temperature (1950−1990; Fig. 5a red) for a particular day of the year (1−365); $T_{d,y}$ is the temperature for a particular day of the year $d$ and a year $y$; $p0_c$ is the prevalence of OSA for a given country $c$; and $APOP_{c,y}$ is the given adult population for a country $c$ in year $y$; $RLE_c$ is the average years of life lost for a given fatal motor vehicle accident for a country $c$, also extracted from the GBD[73]. Uncertainty intervals for YLLs were constructed using an $RR_{MVA\_OSA}$ of 50% (lower bound) and 150% (upper bound), using 100% as the mean.

Similarly, $YLD_{c,y}$ can be calculated as follows:

$$YLD_{c,y} = \frac{1}{365}\sum_{d=1}^{365}[RR_c(T_{d,y}) - RR_c(Thist_{d,1950−90})]*p0_c*APOP_{c,y}*DW*L$$

(3)

where DW is the daily disability weight associated with OSA, and was estimated to be between 0.05−0.11 based on previous literature[22,72]. Uncertainty intervals for YLDs were constructed using a disability weight of 0.05 (lower bound) and 0.11 (upper bound), using 0.08 as the mean. L is the duration of the "diseases/impairment" associated with OSA on a given day. Here, we conservatively assumed this duration to be 1 day, similar to a previous study on quality of life lost to heatwaves in Japan[77]. Other terms are similar to the calculation of $YLL_{c,y}$. Finally, $DALY_{c,y}$ can be calculated as follows:

$$DALY_{c,y} = YLL_{c,y} + YLD_{c,y}$$

(4)

The economic cost of wellbeing loss can be calculated using estimated values of DALYs. Here we used values of one time (lower bound), two times (mean) and three times (upper bound) GDP per capita (2023 USD) as proxies for the value of a DALY, as used in previous studies[78,79]. In our primary model, to isolate the effect of temperature on DALY modelling, we fixed the prevalence of OSA and population per country to a constant number (2023 estimate) until 2100. However, in a

sensitivity analysis, we account for these factors using population forecasts from the UN world prospect 2024 and using projection on increased OSA prevalence (30% increase by 2050) due to increased obesity[34].

## Workplace productivity loss modelling

Sleep disorders have a considerable impact on productivity and are associated with increased likelihood of absenteeism and presenteeism (being at work but not functioning at full capacity)[80]. On average, people with OSA report five additional days of sick leave per year, relative to people without OSA, and spend 6.8% of their working days with reduced productivity[22,35]. The additional absenteeism and presenteeism due to the increase in OSA prevalence days were quantified using similar methodology to the wellbeing burden in the previous section. The overall labor loss was then calculated by adding absenteeism and presenteeism days and multiplying by the average gross domestic product per capita employed. Presenteeism and absenteeism days for a given country $c$ and year $y$ can be calculated as follows:

$$PRES_{c,y} = \frac{NWD}{365}\sum_{d=0}^{365}[RR_c(T_{d,y}) - RR_c(Thist_{d,1950−90})]$$
$$*p0_c*LF_c*PRES_{rate}, \text{ and}$$

(5)

$$ABS_{c,y} = \frac{NWD}{365}\sum_{d=0}^{365}[RR_c(T_{d,y}) - RR_c(Thist_{d,1950−90})]*p0_c*LF_c*ABS_{rate}$$

(6)

where $NWD$ is the number of working days per person on a given year for a full-time employee, here assumed to be 235 (assuming 5 weeks of annual leave). $PRES_{rate}$ and $ABS_{rate}$ is the daily absenteeism on presenteeism rate (2.1% and 6.8% for absenteeism and presenteeism, respectively). $LF_c$ is the labor force for a given country $c$ in 2023, downloaded from the World Bank. Since labor force estimates do not account for part-time labor, we weighted overall labor force using estimate of part-time worker percentage from the World Bank. We multiplied labor force population by half the percentage of part time

employment (assuming part-time employment is 50% full time equivalent) and subtracted it to the overall labor force. The overall labor loss was then calculated as follows:

$$LL_{c,y} = (PRES_{c,y} + ABS_{c,y}) * \frac{GDP \ per \ capita \ employed}{365} \qquad (7)$$

## Reporting summary

Further information on research design is available in the Nature Portfolio Reporting Summary linked to this article.

## Data availability

The dataset associated with this study is stored in a proprietary repository (Withings) and cannot be shared publicly due to concern for privacy, ethical and legal reasons. The investigator team accessed the data through an application process to Withings, and a formal data sharing agreement designed to safeguard user confidentiality, as outlined in the terms and conditions and privacy policy documentation. Queries for data access can be directed to Withings (data_compliance@withings.com) with a timeframe for response of four weeks. Specific de-identified raw data that support the findings of this study, including individual data, are available from the corresponding author (bastien.lechat@flinders.edu.au) upon request subject to ethical and data custodian (Withings) approval described above. The timeframe for response to requests will be up to four weeks. ERA5 weather data and climate model projections are freely available from the Copernicus data store (https://cds.climate.copernicus.eu/). Information on country GDP, rate of motor vehicle accident, and population number are freely available from the world bank (https://www.worldbank.org/ext/en/home), the UN world population prospect (https://population.un.org/wpp/), and the GBD. Exposure-response curve between temperature and OSA RR derived in this study are available on GitHub: https://github.com/bastienlechat/OSA_climate. Wellbeing and economic projection modelling for each country is also available on GitHub: https://github.com/bastienlechat/OSA_climate. Source data are provided with this paper.

## Code availability

Code to reproduce the figures, wellbeing projections and economic projections are all available on GitHub: https://github.com/bastienlechat/OSA_climate.

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

## Acknowledgements
D.J.E. is supported by a National Health and Medical Research Council (NHMRC) of Australia Leadership Fellowship (1196261). B.L. is supported by a NHMRC of Australia Emerging Leadership Fellowship (2025886). J.L.P. and S.B. are partly supported by the French National Research Agency in the framework of the "Investissements d'avenir" program (ANR-15-IDEX-02) and the "e-health and integrated care and trajectories medicine and MIAI artificial intelligence" Chairs of excellence from the Grenoble Alpes University Foundation; MIAI @ Grenoble Alpes, (ANR-19-P3IA-0003). The results contain modified Copernicus Climate Change Service information 2020. Neither the European Commission nor ECMWF is responsible for any use that may be made of the Copernicus information or data it contains. We acknowledge the World Climate Research Programme, which, through its Working Group on Coupled Modelling, coordinated and promoted CMIP6. We thank the climate modelling groups for producing and making available their model output, the Earth System Grid Federation (ESGF) for archiving the data and providing access, and the multiple funding agencies who support CMIP6 and ESGF.

## Author contributions
B.L., J.M., L.P., B.T., B.K. and D.J.E. developed the study concepts and aims. B.L., D.P.N. and J.M. performed the data extraction. B.L., B.T., D.P.N., J.M., A.C.R. and B.K. contributed to the data analysis. B.L., J.M., L.P., D.P.N., and D.J.E. drafted the manuscript. B.L., J.M. L.P., A.C.R., H.S., D.V., S.B., J.F., B.T., B.K., R.A., J.L.P., P.E., D.J.E. provided important insight on data analysis, interpretation and contributed to drafting and to the final version of the manuscript. All authors had full access to all the data in the study and accept responsibility to submit for publication.

## Competing interests
P.E. serves as a consultant for Withings. Outside the submitted work, B.L. has had research grants from Withings, Medical Research Future Fund and NHMRC. Outside the submitted work, D.J.E. has had research grants from Bayer, Apnimed, Takeda, Invicta Medical, Eli Lilly and Withings. D.J.E. currently serves as a scientific advisor/consultant for Apnimed, Invicta Medical, Takeda, SleepRes and Mosanna. A.C.R. has received research funding from the Lifetime Support Authority, Sleep Health Foundation, Flinders Foundation, Medical Research Future Fund, NHMRC, the Hospital Research Foundation, Compumedics, and Sydney Trains, and speaker and consultancy fees from Teva Pharmaceuticals, Sealy Australia, and the Sleep Health Foundation for work unrelated to this study. H.S. reports consultancy and/or research support from Re-Time Pty Ltd, Compumedics Ltd, the American Academy of Sleep Medicine Foundation, and Flinders University. R.J.A. reports research support from the NHMRC, Flinders Foundation, the Hospital Research Foundation, Big Health, Philips Respironics, ResMed Foundation, Flinders University, Sydney Trains and, and speaker and consultancy fees from SomnoMed. P.C. reports grants from NHMRC, Medical Research Future Fund, Flinders Foundation, Invicta Medical, Garnett Passe and Rodney Williams Memorial Foundation, Defence Science and Technology Group. None of the other authors have any potential conflicts to declare.
