## [Peer Review File · Nature Communications]

Global warming may increase the burden of obstructive sleep apnea.

Corresponding Author: Dr Bastien Lechat

Version 0:

Reviewer comments:

Reviewer #1

(Remarks to the Author)

This study investigates the relationship between environmental temperature and obstructive sleep apnea by analyzing large-scale user data. Overall, the study is well-designed and holds significant implications, but I have several comments that should be considered to address the worth of investigation. My comments follow in order of manuscript flow.

1. Line 30: Currently, the text described $AHI \geq 15$ as simply "having OSA" is abstract and may cause confusion. Consider revising for clarity to distinguish between mild, moderate, and severe OSA.
2. Line 28: The text states 125,295 users initially, but the final number of participants analyzed is 116,200. The abstract should reflect this discrepancy accurately by clearly stating the final sample size included in the analysis.
3. Lines 55–56: The economic burden dimension is only listed for Australia and the United States. Is that because the study focused on data from those two countries?
4. Lines 69–70: The description of the FDA-cleared under-mattress sleep sensor lacks references.
5. The paper does not explain how participants were recruited, nor how eligibility for the study was determined. This information should be included in detail.
6. Line 169: the study reports that females make up 22.7% of participants. If this study explores the prevalence of OSA in the population based on this database, I think the authors should be careful with their language and clarify its representativeness
7. In the current analysis, meteorological factors only consider cloud cover and relative humidity. How about wind speed and precipitation?
8. For the CMIP6 data, did the authors apply historical data (ERA5) adjustments for statistical downscaling corrections? Were Taylor diagrams used to evaluate the performance of the models selected?
9. Where does the population data used in future projections come from? Do these projections account for potential changes in population demographics or adaptation to temperature variations?
10. I find the author's description of OSA prevalence is inconsistent with the results. While the overall prevalence of moderate-to-severe OSA (25.4%) and severe OSA (8.9%) is reported as population percentages, the study's outcomes are derived using an individual-level analysis framework under the CTS approach, which examines whether $AHI \geq 15$ for each individual each night. The author should explicitly clarify how OSA prevalence is defined and assessed. Furthermore, the statement in the introduction (Lines 60–62) that "no research has quantified changes in prevalence" is overly definitive, especially considering prior studies that have explored similar data processing and analysis methods.

Reviewer #2

(Remarks to the Author)

Thank you for asking me to review this interesting study that aims at understanding the effect of temperature on OSA severity with particular regards to global warming hypothesizing that the further increase in temperatures could lead to increased OSA burden.

The study is novel and the findings are of interest particularly for policy makers. The most important limitations have been addressed by the authors particularly with regards to the lack of data on comorbidities and OSA features but also regarding the diagnosis of OSA that didn't derive from polygraphies or polysomnographies, the current gold standard for OSA

diagnosis.

I have some comments that I would like the Authors to address before considering this manuscript for publication.

1. As weather information were time-matched for every user based on their closest location it would be useful to have an idea on how precise this location estimate was. In other words, can the author calculate the distance between the real user location and the point that was considered for temperature evaluation and possibly consider this variable as potential confounder?
2. Furthermore, the results seem biased by the fact that patients from low latitude countries such as Africa and South America were underrepresented. This can also affect generalizability of the findings.
3. When looking at OSA prevalence worldwide in the study by Benjafield AV and coauthors (10.1016/S2213-2600(19)30198-5) it seems that the overall picture slightly differs from the nearable-derived OSA prevalence of the present study. How do the authors explain this difference?
4. The authors rightly considered PM 2.5 as indicator of air quality. There is also evidence that air pollution might contribute to worsen OSA (<https://doi.org/10.1164/rccm.200912-1797OC>) in studies conducted in the US although such association was not seen in European patients (10.1007/s11325-023-02918-w). Thus, it could be possible that PM 2.5. contributed as risk factor more than the temperature in relation with OSA severity. This merits to be discussed and addressed.
5. On the same point, most evidence relies on the effect of PM10 on OSA severity but this pollutant was not considered in the present study making the evaluation of air quality somewhat limited.
6. In addition to the previous two points as indoor air pollution levels are typically 2-5 times higher than outdoor pollution levels the authors should discuss this limitation given that OSA severity assessment was performed indoor.
7. The calculation of the global warming-related increase in OSA prevalence is an interesting approach. However, if I understand well it does not consider other important determinants of OSA other than temperature. In particular, I believe that not considering obesity in this projection makes the findings difficult to interpret.

Reviewer #3

(Remarks to the Author)

The paper addresses an important and timely public health concern, exploring the association between ambient temperature and obstructive sleep apnea (OSA) prevalence, while projecting potential health and economic burdens related to global warming through 2100. The paper contributes to the growing body of literature on climate-related health risks, and I appreciate the effort invested in compiling such a large dataset. Below please find my feedback.

- The manuscript makes strong causal claims linking ambient temperature increases to OSA prevalence, DALYs, and productivity losses. Given the observational nature of your study, it would be helpful to frame these associations more cautiously. The assumed causal pathway (Global Warming -> OSA -> Disability/Death -> Economic Loss) could be influenced by unaddressed confounding factors.
- BMI and obesity trends are critical confounders that could independently affect OSA prevalence. Since obesity rates have been rising globally, they may confound the observed temperature-OSA relationship. Addressing these confounders through appropriate adjustments or sensitivity analyses could strengthen the validity of your findings.
- Excluding nights with less than 5 hours of sleep could introduce bias, especially if shorter sleep duration correlates with OSA severity. Clarifying how missing data were handled and considering the potential impacts of selection bias would be valuable.
- It appears the outcome is daily OSA status, but the term "OSA prevalence" is used throughout. Clarifying this distinction would improve clarity.
- The method for comparing current and historical temperatures (1950-1990) needs more detail. Including model specifications and adjustment variables in the supplementary materials would be helpful.
- Relying solely on odds ratios can overstate effect sizes, particularly with a moderate OSA prevalence (about 25%). For example, in Figure 1b, a 30% increase in odds from 1.3 to 1.69 translates to only about a 6% increase in actual risk (if no other adjustments are made for this conversion). Presenting absolute risk differences or risk ratios could provide a clearer public health context.
- The presentation of the results between the 99th percentile (27.3°C) and the 25th percentile (6.4°C) reflects a >20°C difference, which seems disproportionate when assessing the impact of global warming. Considering that projected temperature increases are around 1.5°C to 3°C by 2100, using more realistic temperature contrasts could improve the relevance of your findings.

Version 1:

Reviewer comments:

Reviewer #1

(Remarks to the Author)

The authors have adequately addressed my concerns. I have no further comments.

Reviewer #2

(Remarks to the Author)

I believe the manuscript has greatly improved after the reviewers' suggestions.

No further comments from my end

Reviewer #3

(Remarks to the Author)

No further comments.

We thank the reviewers for their appraisal of our manuscript and helpful suggestions for improvement. We have revised the manuscript accordingly and responded to each of the queries below.

Reviewer #1 (Remarks to the Author):

This study investigates the relationship between environmental temperature and obstructive sleep apnea by analyzing large-scale user data. Overall, the study is well-designed and holds significant implications, but I have several comments that should be considered to address the worth of investigation. My comments follow in order of manuscript flow.

1. Line 30: Currently, the text described $AHI \geq 15$ as simply "having OSA" is abstract and may cause confusion. Consider revising for clarity to distinguish between mild, moderate, and severe OSA.

Response: We unfortunately cannot go into more detail regarding OSA definition in the abstract due to the space constraints (150-word limits in the abstract). We have however, clarified the definition of OSA and severity categories in the main manuscript to improve clarity (see response to query 10 below).

2. Line 28: The text states 125,295 users initially, but the final number of participants analyzed is 116,200. The abstract should reflect this discrepancy accurately by clearly stating the final sample size included in the analysis.

Response: We have now removed the 125,295 number from the abstract and only included the final number of participants included in the analysis - 116,620. We unfortunately cannot go into more detail regarding inclusion/exclusion criteria and number of participants excluded from the final analysis due to the space constraints (150-word limits in the abstract).

3. Lines 55–56: The economic burden dimension is only listed for Australia and the United States. Is that because the study focused on data from those two countries?

Response: To the best of our knowledge, there is no published assessment of workplace productivity and wellbeing loss from OSA in other countries. We have clarified this as follow: “This is estimated to cost the Australian economy nearly 4 billion AUD²² and the US economy nearly \$87 billion US per year²³; while data for other countries is lacking”. In our study, we estimated the burden in all available countries (29 in total).

4. Lines 69–70: The description of the FDA-cleared under-mattress sleep sensor lacks references.

Response: We have added the 510(k) number, see in the “Participant” section of the method: “(Withings Sleep Analyzer/Sleep Rx; WSA; FDA 510(k) number: K231667)”

5. The paper does not explain how participants were recruited, nor how eligibility for the study was determined. This information should be included in detail.

Response: We have expanded the “Participants section” in the methods to address the reviewers’ suggestion as follow:

“Retrospective data were acquired from 125,295 participants who registered to use an FDA-cleared under-mattress sleep sensor (Withings Sleep Analyzer/Sleep Rx; WSA; FDA 510(k) number: K231667) between January 2020 and September 2023. To be included in this analysis, participants were required to have ≥ 28 nights of data and an average of ≥ 4 sleep recordings per week. When participants sign-up to the Withings app, they are prompted to fill in their age, sex, height, and weight. Participants were required to be 18 years or older to be included in this study. Participants were geo-localized to the closest city within a given time zone. More precise location was not available due to concern for re-identification and privacy. All participants provided written consent through the

Withings app for their deidentified data to be used for research purposes when signing up for a Withings account. The current study was approved by the Flinders University Human Research Ethics Committee (Project number: 4291).”

6. Line 169: the study reports that females make up 22.7% of participants. If this study explores the prevalence of OSA in the population based on this database, I think the authors should be careful with their language and clarify its representativeness.

Response: We wholeheartedly agree and have modified our discussion as follow: “In addition, 77% of the participants in the current study were male. We did not detect a large difference between sex in the temperature-OSA association. However, this study sample may not be representative of the wider general population and does not resolve the long-standing issue in sleep research of under-representation of women in such analyses.”

7. In the current analysis, meteorological factors only consider cloud cover and relative humidity. How about wind speed and precipitation?

Response: This is an interesting suggestion. Accordingly, we have updated all models to incorporate 24h mean wind speed and total precipitation. In the methods, in statistical analysis, the model is now described as follows: ‘We used participant/year/month strata intercepts, natural splines of time (day of the year; with 4 degrees of freedom [df]), and indicators of the day of the week to model individual baseline risks, in addition to shared long-term, seasonal, and weekly trends. Models were further adjusted for total cloud cover (4 df), relative humidity (4 df), density of particulate matter with aerodynamic diameter <2.5 $\mu\text{g}/\text{m}^3$ (4 df), surface pressure (4 df), total precipitation (4 df), wind speed (4 df) and variation in total sleep time (4 df) as these potential confounders may have effects on OSA prevalence within the 1-month stratum²⁷.’

Existing and new models were compared in a subsample with minimal changes, per the below figure. Updated models including wind speed and precipitation have been reported in the amended work

*note: this analysis was conducted in a subset (N=20,000) of users to reduce computational load.

8. For the CMIP6 data, did the authors apply historical data (ERA5) adjustments for statistical downscaling corrections? Were Taylor diagrams used to evaluate the performance of the models selected?

Response: Thank you for highlighting this omission. We now detail the processing of the CMIP6 data more thoroughly in the supplementary material including how we bias corrected the data using linear scaling (see ref below): “We used the climate scenarios CMIP6¹⁰ from the 2021 sixth assessment report of the IPCC11 – called Shared Socioeconomic Pathway (SSP). We downloaded daily temperatures from 2023 to 2100 projections for 4 SSPs (SSP126, SSP245, SSP370, SSP585), based on 27 climate models from multiple countries and climate modelling groups. 4 models were excluded due to missing historical baseline data. For the remaining 23 models, we extracted the daily average temperature at 2m from the ground for each of the main cities (square of 500x500km around the location). Each model was bias corrected to the ERA5 dataset and rescaled using linear scaling correction. Averaged daily projections across models were then calculated as the daily median of the 23 projections and the ensemble median was used for all wellbeing and economics modelling.” Model validation was done qualitatively by comparing the single trace and ensemble average of all CMIP6 models with the ERA5 data. See example below for Paris– light grey traces are bias corrected single CMIP6 projections and black is the ensemble average of all CMIP6 projections and blue is ERA5 dataset. Since we used the ensemble average for CMIP6 projections, the standard deviation was not comparable to ERA5 models, so we did not use Taylors diagrams.

Maraun, D. (2016). "Bias Correcting Climate Change Simulations - a Critical Review." Current Climate Change Reports 2(4): 211-220.

9. Where does the population data used in future projections come from? Do these projections account for potential changes in population demographics or adaptation to temperature variations?

Response: The population estimates were extracted from the United Nations data from the 2022 World Populations prospect reports, see here: <https://population.un.org/wpp/>. This information was provided in the supplementary material which we have now decided to move to the main manuscript in the methods to be more accessible (see “Country level information” section in the methods). In

addition, the 2024 estimates have been published since the last round of analysis, so we updated our results with the new estimates.

For the main analysis, population was fixed at the 2023 adult population. For a sensitivity analysis, we now use projections from the prospect reports up until 2100, in addition to increased rate of OSA due to potential increase in BMI and obesity. See response to **reviewer 2 suggestions #7** for the additional sensitivity model.

The reviewer is also right that our projections do not account for future adaptation to temperature variations. This is because there are limited studies that provide evidence of the effect of heat adaptation strategies on sleep and OSA (see below article below for a recent review). We now provide a more balance discussion on heat adaptation in the discussion section of the article: "... We also assumed constant heat adaptation at the population level. However, heat-related mortality in Europe has declined over the past two decades⁷⁷, and global air conditioning adoption has increased from 19.3% (2000) to 35.3% (2021). Studies that have investigated heat adaptation strategies and its effects on sleep and OSA are scarce¹⁵. Thus, we could not account for heat adaptation. In addition, heat adaptation is not uniform: highly developed countries have 44-48% household air conditioning coverage, while low and medium developed countries have 4% and 14%, respectively⁷⁸. Thus, given our sample's socioeconomic bias toward developed nations, we likely underestimate the global impact of temperature-induced OSA, even without accounting for increased adaptation capabilities."

Chevance, G., et al. (2024). "A systematic review of ambient heat and sleep in a warming climate." Sleep Medicine Reviews.

10. I find the author's description of OSA prevalence is inconsistent with the results. While the overall prevalence of moderate-to-severe OSA (25.4%) and severe OSA (8.9%) is reported as population percentages, the study's outcomes are derived using an individual-level analysis framework under the CTS approach, which examines whether $AHI \geq 15$ for each individual each night. The author should explicitly clarify how OSA prevalence is defined and assessed.

Response: This point was also raised by other reviewers. Accordingly, we have clarified the terminology throughout the manuscript. In the results we now employ "OSA prevalence" for country estimates only, and we use the word "probability of nightly OSA" or "nightly severe OSA" for the CTS outcomes. We have also clarified the terminology in the method section as follow: "Our primary outcome of interest was the nightly OSA status for a given participant on a given night, based on the AHI and clinically defined severity categories (nightly OSA: $AHI \geq 15$; and nightly severe OSA: $AHI \geq 30$ event/h). For each participant, we also calculated the average OSA severity category based on the average AHI on all available nights and using pre-defined clinical cut-offs (mild: 5-15; moderate: 15-30 and severe: ≥ 30 events/h). The prevalence of OSA per country was calculated based on the average OSA severity category for each participant."

In addition, we have also moved most of the methods on calculations of DALYs and increased OSA prevalence from the supplementary to the main manuscript – this should make the calculation of OSA-person days and increased OSA prevalence due to temperature clearer. Specifically, the increase in OSA prevalence from temperature is calculated from this part of the equation:

$$\frac{1}{365} \sum_{d=0}^{365} [RR_c(T_{d,y}) - RR_c(Thist_{d,1950-90})] * p0_c$$

11. Furthermore, the statement in the introduction (Lines 60–62) that "no research has quantified changes in prevalence" is overly definitive, especially considering prior studies that have explored similar data processing and analysis methods.

Response: We have reworded the sentence as follow: “Furthermore, to the best of our knowledge, personal and social burden of OSA due to increased temperatures, including changes in prevalence, wellbeing burden, and workplace productivity loss have not been quantified to date.”

Reviewer #2 (Remarks to the Author):

Thank you for asking me to review this interesting study that aims at understanding the effect of temperature on OSA severity with particular regards to global warming hypothesising that the further increase in temperatures could lead to increased OSA burden.

The study is novel and the findings are of interest particularly for policy makers. The most important limitations have been addressed by the authors particularly with regards to the lack of data on comorbidities and OSA features but also regarding the diagnosis of OSA that didn't derive from polygraphies or polysomnographies, the current gold standard for OSA diagnosis. I have some comments that I would like the Authors to address before considering this manuscript for publication.

1. As weather information were time-matched for every user based on their closest location it would be useful to have an idea on how precise this location estimate was. In other words, can the author calculate the distance between the real user location and the point that was considered for temperature evaluation and possibly consider this variable as potential confounder?

Response: Unfortunately, the investigator team was not able to access the users' residential address due to for the potential for re-identification and privacy concerns. We have clarified this point in the methods, under the participants heading, as follows:

“More precise location was not available due to concern for re-identification and privacy.”

2. Furthermore, the results seem biased by the fact that patients from low latitude countries such as Africa and South America were underrepresented. This can also affect generalizability of the findings.

Response: We agree with reviewer that this is a limitation. Unfortunately, this is a limitation in the sleep research field in general – there is very little objective population data available for LMICs. We discuss this point as follows in the discussion section: “There are large global inequalities in availability of population sleep health data – with only 22% WHO member states (mostly highly developed) having published population data in sleep duration⁶⁸. Similarly, published global estimates of OSA prevalence are derived from population studies in mostly high socio-economic countries¹⁹. Our study is also biased towards higher socioeconomic-status countries and highlights the urgent need for global strategies to collect appropriate sleep and temperature data worldwide⁶⁸”

In addition, we provide context for the interpretation of our health economics model output given the lack of lower income countries in our dataset:

“We also assumed constant heat adaptation at the population level. However, heat-related mortality in Europe has declined over the past two decades⁷⁷, and global air conditioning adoption has increased from 19.3% (2000) to 35.3% (2021). Studies that have investigated heat adaptation strategies and its effects on sleep and OSA are scarce¹⁵. Thus, we could not account for heat adaptation. In addition, heat adaptation is not uniform: highly developed countries have 44-48% household air conditioning coverage, while low and medium developed countries have 4% and 14%, respectively⁷⁸. Thus, given our sample's socioeconomic bias toward developed nations, we likely underestimate the global impact of temperature-induced OSA, even without accounting for increased adaptation capabilities.”

3. When looking at OSA prevalence worldwide in the study by Benjafeld AV and coauthors (10.1016/S2213-2600(19)30198-5) it seems that the overall picture slightly differs from the nearable-derived OSA prevalence of the present study. How do the authors explain this difference?

Response: In our previous work, we directly compared our OSA prevalence estimates with the study from Benjafield (see paper below). We found that for most countries – the prevalence of moderate to severe OSA is similar to the Benjafield paper (see Table 1 in the paper). Interestingly, Benjafield prevalence estimates range from 3 to 40%, and are very inconsistent even amongst neighbouring countries (e.g., France is 36.6%, but Belgium is 15.7%). This is likely due Benjafield et al.'s variable methodology that relied on a literature search and not on country-by-country objective measurement of OSA burden. They used 16 population-based studies and extrapolation to the remaining countries based on matched demographics and the associated limitations of single-night testing.

Our prevalence estimates using the WSA are however more consistent across countries (range: 15 to 29%), likely due to the standardized nature of our methodology. On the other hand, there is a selection bias in our study, with more men, which could inflate the prevalence. The people also self-selected to buy the device, potentially due to concerns about their sleep, so it may not be representative of the general population, and may lead to noise in the estimation of OSA prevalence (we have mentioned these limitations in our discussion).

Lechat, B., et al. (2022). "Multinight Prevalence, Variability, and Diagnostic Misclassification of Obstructive Sleep Apnea." *Am J Respir Crit Care Med* **205**(5): 563-569.

4. The authors rightly considered PM 2.5 as indicator of air quality. There is also evidence that air pollution might contribute to worsen OSA (<https://doi.org/10.1164/rccm.200912-1797OC>) in studies conducted in the US although such association was not seen in European patients (10.1007/s11325-023-02918-w). Thus, it could be possible that PM 2.5. contributed as risk factor more than the temperature in relation with OSA severity. This merits to be discussed and addressed.

Response: We appreciate the reviewer feedback and insights into the potential contribution of air pollution into our findings here and in the next 2 suggestions. We have extended our analysis to investigate this more thoroughly. Firstly, to address whether PM 2.5. contributed as risk factor more than the temperature, we have run three different models. Model 1 adjusted for all confounders except air pollution, model 2 adjusted for model 1 + PM 2.5, and model 3, adjusted for model 1 + PM10. In all models, the associations between temperature and nightly OSA probability was similar – that suggests that neither PM2.5 or PM10 were major confounders in our study. We also looked at an interaction between air pollution (PM2.5) and temperature, since air pollution has been shown to modify the association between high temperature and mortality (see article below). However, we found no significant interactions in our study.

We have now included this new analysis in the results section (including a new supplementary Figure S7) as follows: "The exposure-response curve between ambient temperature and the probability of nightly OSA also remained similar when adjusting for particulate matter with aerodynamic diameter <10µm (Figure S7), previously shown to be associated with increased OSA severity in some studies²⁶. The association between temperature and nightly OSA was not modified by air pollution (Figure S7)."

Rai, M., et al. (2023). "Heat-related cardiorespiratory mortality: Effect modification by air pollution across 482 cities from 24 countries." *Environ Int* **174**: 107825.

5. On the same point, most evidence relies on the effect of PM10 on OSA severity but this pollutant was not considered in the present study making the evaluation of air quality somewhat limited.

Response: See our response above where we address this point.

6. In addition to the previous two points as indoor air pollution levels are typically 2-5 times higher than outdoor pollution levels the authors should discuss this limitation given that OSA severity assessment was performed indoor.

Response: We have added this limitation in the discussion paragraph regarding air pollution, temperature and nightly OSA associations: “The potential confounding effect of air pollution on the association between high temperature and nightly OSA status may also have been underestimated since we lacked precise geographical locations and indoor measurements of air pollution.”

7. The calculation of the global warming-related increase in OSA prevalence is an interesting approach. However, if I understand well it does not consider other important determinants of OSA other than temperature. In particular, I believe that not considering obesity in this projection makes the findings difficult to interpret.

Response: We wanted to isolate the effect of rising temperature's impact on OSA-related wellbeing and productivity by holding other factors constant. Hence, we did not adjust for anything else (such a population growth, change in OSA prevalence rates, adaptation, etc).

Increased OSA prevalence due to increase obesity (~30% increase in moderate to severe prevalence by 2050, in the US [1]) inflates our estimates, and we are worried that these estimates will be interpreted as being “solely” due to rising temperatures. Thus, we have elected to keep this as a supplementary analysis. In the methods, the analysis is described as follows: “In our primary model, to isolate the effect of temperature on DALY modelling, we fixed the prevalence of OSA and population per country to a constant number (2023 estimate) until 2100. However, in a sensitivity analysis, we account for these factors using population forecasts from the UN world prospect 2024 and using projection on increased OSA prevalence (30% increase by 2050) due to increase obesity⁶².”

In the results, the new sensitivity analysis is described as follows (and inclusion of new Table S7 and S8): “Accounting for population forecast and potential 30% increase in OSA prevalence by 2050 due to rise in obesity⁶² would further increase these estimates in most countries by 10 to 25%. Comparison of the main model estimates vs the alternative scenario is available in Table S7 and S8. Under these assumptions, any other scenarios than SSP126 would be estimated to incur an additional to 16.9 to 47.4 million DALYs lost over the 2023 to 2100 period.”

Finally, we have also added a discussion surrounding the limitations of our economic modelling, including rising BMI rates and other limitations raised by reviewers, in the discussion as follows: “Our main analysis aimed to model rising temperature's impact on OSA-related wellbeing and productivity by holding other factors constant. However, BMI is projected to increase over the next 50 years, likely leading to higher OSA prevalence⁶³. This suggests our main estimates may understate the future temperature-related OSA burden. Indeed, once accounting for BMI-related increase in OSA prevalence in our sensitivity analyses, our estimates were 10 to 30% higher. On the other hand, recent developments in OSA treatment⁷⁴, including FDA-approved weight loss medications⁷⁵, could help manage OSA more effectively. Given that approximately 80% of OSA patients are currently undiagnosed or untreated⁷⁶, improved treatment rates could serve as an adaptation mechanism to offset temperature-induced increases in OSA burden. We also assumed constant heat adaptation at the population level. However, heat-related mortality in Europe has declined over the past two decades⁷⁷, and global air conditioning adoption has increased from 19.3% (2000) to 35.3% (2021). Studies that have investigated heat adaptation strategies and its effects on sleep and OSA are scarce¹⁵. Thus, we could not account for heat adaptation. In addition, heat adaptation is not uniform: highly developed countries have 44-48% household air conditioning coverage, while low and medium developed countries have 4% and 14%, respectively⁷⁸. Accordingly, given our sample's socioeconomic bias toward developed nations, we likely underestimate the global impact of temperature-induced OSA, even without accounting for increased adaptation capabilities.”

Reviewer #3 (Remarks to the Author):

The paper addresses an important and timely public health concern, exploring the association between ambient temperature and obstructive sleep apnea (OSA) prevalence, while projecting potential health and economic burdens related to global warming through 2100. The paper contributes to the growing body of literature on climate-related health risks, and I appreciate the effort invested in compiling such a large dataset. Below please find my feedback.

1. The manuscript makes strong causal claims linking ambient temperature increases to OSA prevalence, DALYs, and productivity losses. Given the observational nature of your study, it would be helpful to frame these associations more cautiously. The assumed causal pathway (Global Warming -> OSA -> Disability/Death -> Economic Loss) could be influenced by unaddressed confounding factors.

Response: Based on the reviewer's feedback we have revised the text to use more cautious language throughout the manuscript. We have also added a sentence surrounding potential missing confounders in the discussion as follows: "Although our robust time-series analysis design provides support for a potential causal association between high temperature exposure and increased OSA prevalence, other time-varying confounders (e.g., alcohol intake) were not available in the current study, and should be accounted for in future investigations."

2. BMI and obesity trends are critical confounders that could independently affect OSA prevalence. Since obesity rates have been rising globally, they may confound the observed temperature-OSA relationship. Addressing these confounders through appropriate adjustments or sensitivity analyses could strengthen the validity of your findings.

Response: If we understand correctly - there are two parts to this question. Firstly, in our statistical model that investigates the association between temperature and nightly OSA prevalence, BMI and obesity are accounted for by design since the case-time-series design has person-year-month intercept – hence, potential monthly variation in BMI is adjusted for each participant.

Secondly, we agree with the reviewer that our primary economic model does not account for varying BMI rates. This was done by design – since we wanted to isolate the effect of rising temperature's impact on OSA-related wellbeing and productivity by holding other factors constant. Reviewer 2 (suggestions #7) brought up a similar point, which we addressed through an additional sensitivity analysis accounting for increase OSA prevalence due to increase BMI and population growth. Please see our response to **reviewer 2 suggestions 7**.

3. Excluding nights with less than 5 hours of sleep could introduce bias, especially if shorter sleep duration correlates with OSA severity. Clarifying how missing data were handled and considering the potential impacts of selection bias would be valuable.

Response: Missing data in our previous model was removed, this led to ~9.4% of nights excluded. We have clarified this in the results section as follows: "There was 9.4% of nightly recordings missing an AHI recording due to sleep duration below 5 hours.". We also expanded on how much data was excluded due to not enough regular use of the device: "Data for final analysis was acquired from 116,220 users for a combined total of ~62 million nights. The most common reason for excluding participants (N = 8261, ~6.6%) were the lack of regular use of the device (see flow chart S3)."

However, the reviewer raises a good point about missing data potentially introducing bias. Accordingly, we ran an additional sensitivity analysis that looked at the exposure-response curve between temperature and nightly OSA for each quartile of missing data (determined on a per-participant basis) and found that in quartiles with the most missing data due to the 5h cut-offs, the effect tends to be underestimated compared to quartiles with minimum amount of missing data (see Figure below). We now describe these additional findings (and have included a new figure S8) as

follows: “There was a bias towards a lower effect size in the association between temperature and nightly OSA for increased amount of missing data (see Figure S8). Given the association between high temperature and short sleep duration, this may suggest that our estimates are conservative.”

4. It appears the outcome is daily OSA status, but the term "OSA prevalence" is used throughout. Clarifying this distinction would improve clarity.

Response: Thank you, this point was raised by other reviewers, and we have clarified the terminology throughout the manuscript. In the results, “OSA prevalence” is now employed for country estimates only, and we use the terms “probability of nightly OSA” or “nightly severe OSA” for the CTS outcomes. We have also clarified the terminology in the method section as follows: “Our primary outcome of interest was the nightly OSA status for a given participant on a given night, based on the AHI and clinically defined severity categories (moderate-to-severe: $AHI \geq 15$; and severe OSA: $AHI \geq 30$ event/h). For each participant, we also calculated the average OSA severity category based on the average AHI on all available nights and using pre-defined clinical cut-offs (mild: 5-15; moderate: 15-30 and severe: ≥ 30 events/h). The prevalence of OSA per country was calculated based on the average OSA severity category for each participant.”

5. The method for comparing current and historical temperatures (1950-1990) needs more detail. Including model specifications and adjustment variables in the supplementary materials would be helpful.

Response: We have re-written the methods section to improve clarity. Specifically, the calculation of OSA burden for historical vs measured temperatures is now explained in more detail in the methods (including all formulas used). This can be found page 8 and 9.

6. Relying solely on odds ratios can overstate effect sizes, particularly with a moderate OSA prevalence (about 25%). For example, in Figure 1b, a 30% increase in odds from 1.3 to 1.69 translates to only about a 6% increase in actual risk (if no other adjustments are made for this conversion). Presenting absolute risk differences or risk ratios could provide a clearer public health context.

Response: We agree and have changed all results to risk ratios instead of odds ratio.

7. The presentation of the results between the 99th percentile (27.3°C) and the 25th percentile (6.4°C) reflects a $>20^\circ\text{C}$ difference, which seems disproportionate when assessing the impact of global warming. Considering that projected temperature increases are around 1.5°C to 3°C by 2100, using more realistic temperature contrasts could improve the relevance of your findings.

Response: Thank you, we also went back and forth on this issue. Most environmental epidemiological studies on temperature, climate change and health outcomes use the 99th percentiles (3 hottest day in the year) vs the temperature with the lowest risk to quantify exposure-response curves, see (few papers below). Temperature swings of 10 degrees or more are also not uncommon given the increased likelihood of heatwaves with climate change. In addition, 99th vs 25th percentile is different for each country, For example, in Mexico, 25th vs 99th is a temperature range of ~9 degrees – whereas in France it is a temperature range of (24.6 vs 6 degrees); so, it’s difficult to know what a suitable alternative temperature increase would be (e.g. 5 degrees or 10 degrees?). Hence, we have elected to follow the methodology in the field and keep the 99th percentile vs 25th percentile.

1. Luthi, S., et al. (2023). Nat Commun **14**(1): 4894.
2. Martinez-Solanas, E., et al. (2021). Lancet Planet Health **5**(7): e446-e454.
3. Masselot, P., et al. (2023). Lancet Planet Health **7**(4): e271-e281.

Response to additional concerns raised:

1. Unclear handling of missing data

Response: We have clarified how we handle missing data in the response to suggestion 3 from reviewer 3. You can find this page 9. We also included an analysis that quantified the impact of missing data on our estimates, which suggests that our overall estimates are quite conservative – this additional work can be found page 11 of the manuscript, and new figure S8.

2. potential misclassification bias for the exposure as this relies on city-level temperature data to approximate individual exposure

Response: Unfortunately, due to concern surrounding re-identification and privacy, we ethically could not have access users' exact location. This is now clarified on page 5 (see suggestion 1 from reviewer 2) and acknowledged on multiple occasions in the limitation section of the discussion, page 17. While misclassification bias is possible, it is also important to note the lack of precise location is likely to lead to an *underestimation* of the association between daily OSA status and temperature (regression to the mean). As a result, our estimates are conservative.

3. potential misclassification bias as this relies on the under-mattress devices for OSA detection

Response: We agree that the under-mattress device does not measure the contribution of some physiological aspects of OSA (e.g., hypoxia, arousals) which may undermine OSA detection (as mentioned in the discussion). However, validation studies from multiple independent investigations indicate that the WSA is 88% accurate in detecting moderate-to-severe OSA. In addition, multi-night simplified monitoring of OSA (as in this study) has been shown to be a better predictor of health outcomes such as hypertension compared to single night assessment (10.1016/j.chest.2023.01.027). Indeed, conventional single-night gold standard polysomnography can lead to inaccurate OSA diagnoses and severity estimation in 20 to 50% of patients due to high-night-to-night variability of OSA (10.1164/rccm.202107-1761OC). Therefore, the multi-night AHI estimates used in the current study certainly provides comparable or potentially superior insight to polysomnography. We have reworded the discussion to better discuss these points as follows: 'The under-mattress sensor used in this study does not measure the contribution of some physiological aspects of OSA (e.g., hypoxia, arousals) which may undermine OSA detection. However, existing validation studies versus gold standard polysomnography in over 150 participants^{28,31} support the device performance characteristics. Furthermore, OSA prevalence estimates using non-contact multi-night data found in this study yield very similar findings to previously published literature^{19,28}. Similarly, previous studies that used this device have shown that misclassification rates and AHI variability are comparable to other devices³⁰, and that the effect size of the association between the estimated AHI and health outcomes is also similar to existing epidemiological trials^{70,71}. Furthermore, conventional single-night gold standard polysomnography can lead to inaccurate OSA diagnoses and severity estimation in 20 to 50% of patients due to high-night-to-night variability of OSA^{28,29,72}. Thus, these studies provide support that the multi-night AHI estimates used in the current study provide comparable or superior insight to conventional single-night polysomnography AHI while being less cumbersome and allowing nightly data collection over ~1.5 years per individual.'

4. Selection bias due to consumer technology being more readily adopted by individuals or groups with a higher socioeconomic status

Response: We have acknowledged this more thoroughly in responses to **suggestion 2 from reviewer 2** and **suggestion 9 from reviewer 1**. Regrettably, like many areas of health and medicine, selection bias is abundant in the sleep research field broadly, with scarce objective evidence regarding sleep and sleep disorders in lower- and middle-income countries and low socio-economics groups. Given the important influence of the social determinants of health, this may lead to an underestimation of the

global effect of temperature on OSA, since lower- and middle-income countries are less likely to have access to heat mitigation strategies.

As illustrated above, many of the limitations mentioned are likely to result in an *underestimation* of the impact of temperature on the prevalence and burden of OSA, and do not detract from the relevance of, and potential importance of, the novel findings we report here. Despite this potential underestimation, our findings indicate that global warming poses a significant threat to increasing the global OSA burden, with a near doubling in wellbeing loss by the end of the century under the most likely climate scenarios. Even with conservative estimates, our results suggest that the temperature-induced OSA burden (~100 per 100,000 in 2023) are significant compared to the current DALYs rate for high temperature exposure from the Global Burden of Diseases 2021 (180 per 100,000), which only accounts for temperature-induced mortality.